# Vibrational signature of hydrated protons confined in MXene interlayers

Mailis Lounasvuori ®[1], Yangyunli Sun[2], Tyler S. Mathis[3], Ljiljana Puskar[1], Ulrich Schade ®[1], De-En Jiang ®[2,4], Yury Gogotsi ®[3] & Tristan Petit ®[1] ✉

The hydration structure of protons has been studied for decades in bulk water and protonated clusters due to its importance but has remained elusive in planar confined environments. Two-dimensional (2D) transition metal carbides known as MXenes show extreme capacitance in protic electrolytes, which has attracted attention in the energy storage field. We report here that discrete vibrational modes related to protons intercalated in the 2D slits between $Ti_3C_2T_x$ MXene layers can be detected using *operando* infrared spectroscopy. The origin of these modes, not observed for protons in bulk water, is attributed to protons with reduced coordination number in confinement based on Density Functional Theory calculations. This study therefore demonstrates a useful tool for the characterization of chemical species under 2D confinement.

The hydrated excess proton is of fundamental importance in diverse systems ranging from biological processes[1] to catalysis and fuel cells[2]. The solvation structure of a proton is usually described as the Eigen cation[3], where a hydronium ion is solvated by three water molecules, or the Zundel cation[4], where the excess proton is equally solvated by two flanking water molecules. Recent theoretical and experimental studies are increasingly pointing to a distorted structure, or a structure in between these two limiting cases[5]. Due to the dynamic hydrogen-bonding (H-bonding) patterns of water leading to rapidly interconverting proton hydration structures, protons in bulk solution display a broad vibrational signature that spans from ca. $1000 cm^{-1}$ to beyond $3000 cm^{-1}$ [6]. This so-called acid continuum absorption makes it difficult to assign features to any particular structure. However, the infrared (IR) spectra of protonated water clusters present sharper bands compared to the spectra of bulk solutions, which let to breakthroughs in the understanding of proton hydration[7–9]. Recently, ultrafast two-dimensional (2D) IR spectroscopy measurements have demonstrated that protons are mostly solvated as Zundel complexes in bulk solution[10,11]. In contrast to clusters, which are limited to *ca.* 21 water molecules[6], and bulk water, the vibrational modes of hydrated protons in 2D confinement remain largely unexplored.

Water in 2D confinement presents bulk-like water in the planar dimension, while being limited to thicknesses of 1-3 layers of water, thus constituting an ideal platform to bridge the knowledge gap between clusters and bulk water. 2D confinement affects the H-bonding structure of water, leading to anomalous properties such as low dielectric constant[12], exotic phases[13] and high proton conduction[14]. Few reports detailing the spectroscopic characterization of such confined water layers are available to date and are limited to neutral water[15–18]. The hydration structure of protons under confinement requires further attention. Confined protons are indeed expected to play a major role in the pseudocapacitive electrochemical charging of 2D materials such as MXenes in acidic environment[19,20]. MXenes are a large family of 2D transition metal carbides, nitrides and carbonitrides which are most commonly prepared by the selective etching and removal of the A layer from MAX phases[21]. The resulting MXenes are conductive with hydrophilic surfaces terminated by -O, -OH, and other functional groups represented by $T_x$ in the chemical formula. Over the last years, $Ti_3C_2T_x$ MXene has shown remarkable electrochemical properties in acidic electrolytes, putting this material at the forefront of pseudocapacitive energy storage materials[22]. Nevertheless, experimental evidence of the proton intercalation mechanisms in MXene remains elusive so far. Confined water in $Ti_3C_2T_x$ MXene, the most

[1]Helmholtz-Zentrum Berlin für Materialien und Energie GmbH, Berlin, Germany. [2]Department of Chemistry, University of California, Riverside, Riverside, CA, USA. [3]Department of Materials Science and Engineering and A. J. Drexel Nanomaterials Institute, Drexel University, Philadelphia, PA, USA. [4]Present address: Department of Chemical and Biomolecular Engineering, Vanderbilt University, Nashville, TN, USA. ✉e-mail: tristan.petit@helmholtz-berlin.de

studied MXene so far, has been characterized by NMR[23] and inelastic neutron scattering[24]; however, an effective method of directly characterizing the hydrated protons confined within the layers of MXenes has yet to be demonstrated.

In this work, we utilized *operando* Fourier Transform Infrared (FTIR) spectroscopy to probe the water molecules and hydrated protons confined within the interlayer spacing of $Ti_3C_2T_x$ during potential-induced proton intercalation. FTIR spectroscopy is ideally suited for the study of water due to the strong IR absorption of O−H bonds but has not yet been utilized to probe intercalation and solvation phenomena in MXenes. By combining FTIR spectroscopy and theoretical modelling, we have been able to show that the H-bonding structure of hydrated protons electrochemically intercalated into 2D slits between $Ti_3C_2T_x$ MXene layers is fundamentally different from bulk solution.

## Results and discussion
### FTIR spectroscopy analysis of intercalated species in $Ti_3C_2T_x$ MXene interlayer

The experimental setup of the *operando* FTIR system is shown schematically in Fig. 1a (see Supplementary Fig. 1 for a top view of the spectroelectrochemical cell). We use the term *operando* spectroscopy to emphasize the fact that infrared spectroscopic characterization of the $Ti_3C_2T_x$ MXene film was acquired during the CV measurement, without stopping the potential at any fixed value. Each spectrum is averaged over a scanning range of 60 mV (see SI for details). The measurements were made in the attenuated total reflectance (ATR)

mode using microstructured Si substrates as the internal reflection element (IRE). All IR-active vibrational modes of $Ti_3C_2T_x$ MXene itself, except for the terminal O−H stretch, are calculated to appear below the wavenumber range accessible with this setup[25]. The ATR mode helps to remove much of the bulk electrolyte signal since the probing depth is less than the thickness of the $Ti_3C_2T_x$ film (Supplementary Note1, Supplementary Fig. 2). Therefore, the recorded water and hydrated proton signatures only originate from confined water in the interlayer spaces between the MXene sheets and potentially from any bulk-like water present in macropores within the $Ti_3C_2T_x$ film. For these measurements, $Ti_3C_2T_x$ MXene was synthesized according to the method described in[26] (see Methods for details). Successful etching of the parent MAX phase is demonstrated by the X-ray diffraction (XRD) pattern, where only the (00 $l$) reflections are present (Fig. 1b). The sharp (002) reflection at 7.29° corresponds to a *d*-spacing of 12.1 Å and an interlayer distance of 2.7 Å due to residual $Li^+$ and a monolayer of water confined between the layers[27]. The expanded 2D structure of the $Ti_3C_2T_x$ film is evident from the SEM image on a freestanding film.

We first measured the infrared spectrum of the $Ti_3C_2T_x$ film under vacuum before and after exposure to 0.5 M $H_2SO_4$ (Fig. 1c). The presence of water confined in the interlayer spaces of the Li-intercalated $Ti_3C_2T_x$ film is apparent from the absorption features at 1640 cm$^{-1}$ (bending mode) and 3400 cm$^{-1}$ (stretching mode). The sharp band at 3640 cm$^{-1}$ is attributed to dangling O−H bonds of water molecules[6,28]. Exposing the dry $Ti_3C_2T_x$ film to an acidic solution replaces the residual $Li^+$ ions, which were present in the $Ti_3C_2T_x$ interlayer spaces as a result

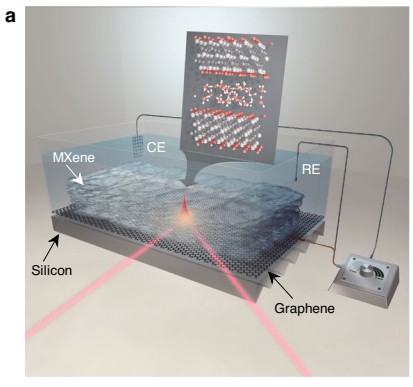

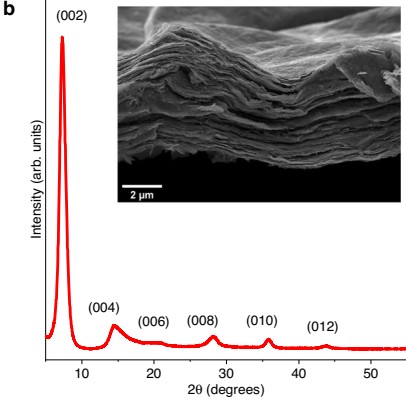

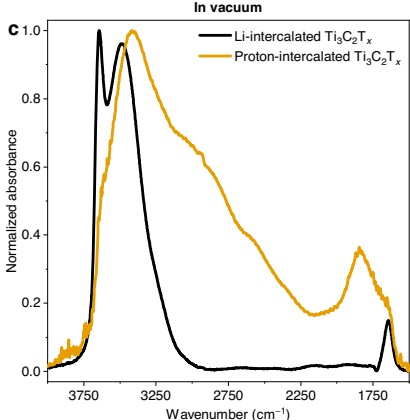

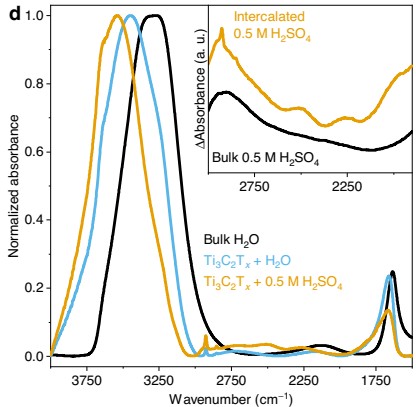

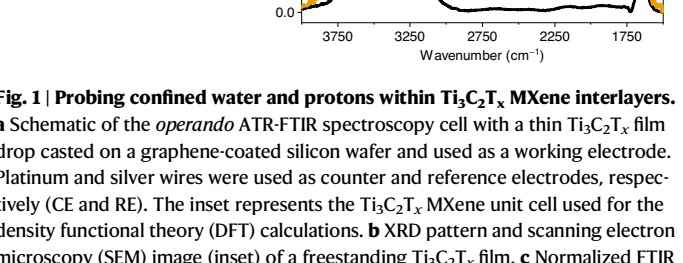

**Fig. 1 | Probing confined water and protons within $Ti_3C_2T_x$ MXene interlayers.** **a** Schematic of the *operando* ATR-FTIR spectroscopy cell with a thin $Ti_3C_2T_x$ film drop casted on a graphene-coated silicon wafer and used as a working electrode. Platinum and silver wires were used as counter and reference electrodes, respectively (CE and RE). The inset represents the $Ti_3C_2T_x$ MXene unit cell used for the density functional theory (DFT) calculations. **b** XRD pattern and scanning electron microscopy (SEM) image (inset) of a freestanding $Ti_3C_2T_x$ film. **c** Normalized FTIR

spectra of $Ti_3C_2T_x$ MXene film before (Li-intercalated) and after (proton-intercalated) exposure to 0.5 M $H_2SO_4$, recorded in vacuum. **d** FTIR spectra of bulk water, confined water in MXene film, and 0.5 M $H_2SO_4$ solution confined in MXene film normalized to highest absorbance. Inset: IR difference of bulk and MXene-confined 0.5 M $H_2SO_4$ solution after subtraction of the spectrum of pure water. The difference spectra are offset for clarity.

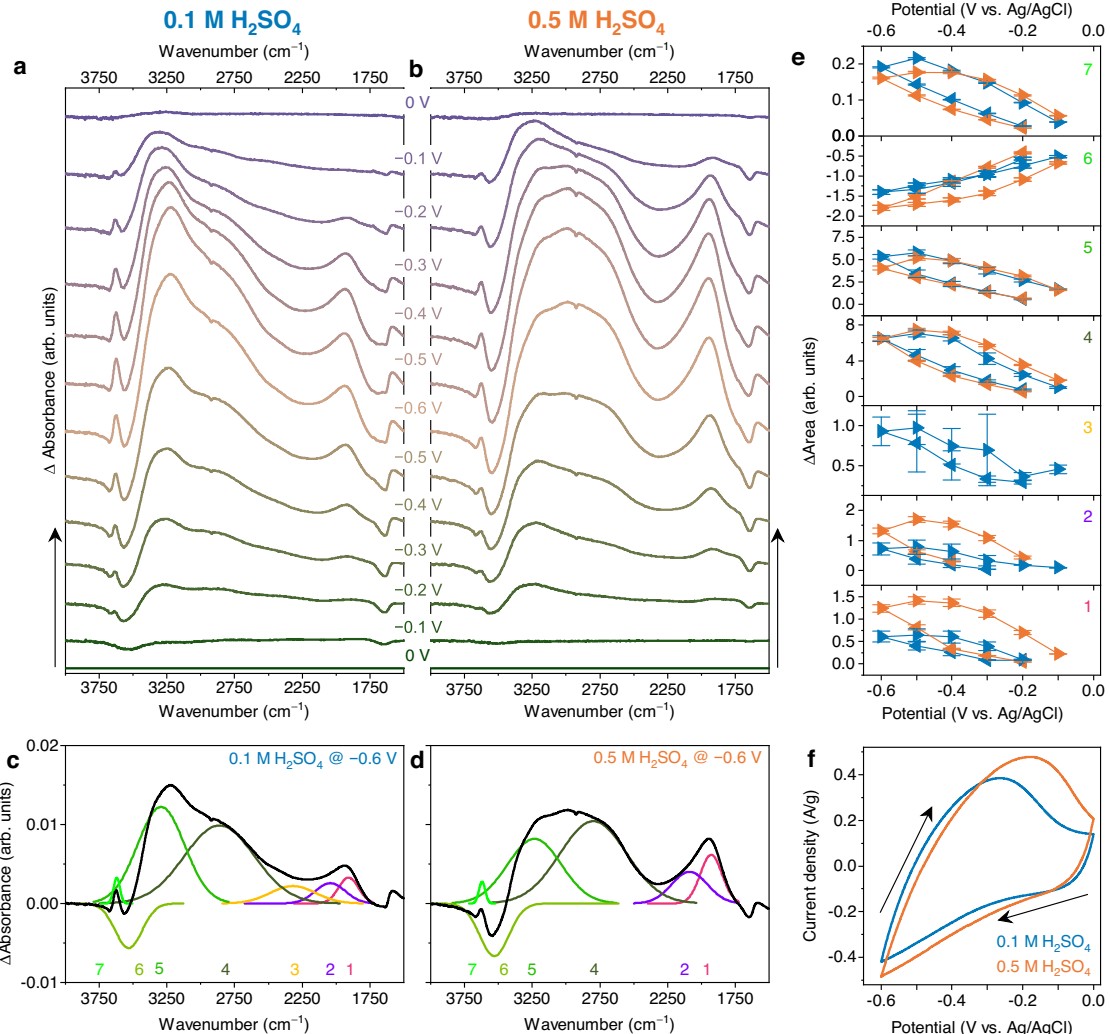

**Fig. 2 | *Operando* electrochemical IR spectroscopy on Ti$_3$C$_2$T$_x$ electrodes.**
**a**, **b** FTIR difference spectra measured during electrochemical cycling in 0.1 M H$_2$SO$_4$ (**a**) and 0.5 M H$_2$SO$_4$ (**b**). The spectrum recorded at 0 V is subtracted from subsequent spectra. Spectra are offset for clarity. **c**, **d** Peak fit of difference spectrum at −600 mV in 0.1 M H$_2$SO$_4$ (**c**) and 0.5 M H$_2$SO$_4$ (**d**). The fit components are numbered from one to seven and the numbers are displayed below the components. Each component was labelled with a different colour for clarity. **e** Integrated peak areas from the fit as a function of potential, with peak numbers corresponding to the numbering in **c**, **d**. **f** CVs recorded during *operando* FTIR measurements.

of the MXene synthesis and delamination process, with protons, leading to a significant decrease in the water bands and the appearance of new bands at 1820 cm$^{-1}$, 2040 cm$^{-1}$, 2560 cm$^{-1}$ and 3090 cm$^{-1}$, which are attributed to confined hydrated protons (see Supplementary Fig. 3 and Supplementary Table 1 for peak fits). The strong reduction of the band at 3640 cm$^{-1}$ indicates that the dangling O−H bonds are to a large extent related to water molecules in the hydration shells of Li$^+$ initially present in the Ti$_3$C$_2$T$_x$ film. The remaining water molecules are most likely bonded in small clusters, as observed in supercritical water[29].

The FTIR spectra for bulk water, water in the initial Ti$_3$C$_2$T$_x$ film and 0.5 M H$_2$SO$_4$ solution intercalated in the Ti$_3$C$_2$T$_x$ film normalized to the highest absorbance are presented in Fig. 1d. The bulk 0.5 M H$_2$SO$_4$ spectrum is nearly identical to the pure water spectrum and therefore not shown here. The O−H stretching band shifts strongly toward higher wavenumbers going from bulk to confined water in the presence of Li$^+$ and further in the confined acid solution after the removal of the intercalated Li$^+$. The water present in the proton-intercalated MXene appears to be more weakly H-bonded compared to the water present before the removal of Li, which may be related to the smaller interlayer spacing found in proton-intercalated Ti$_3$C$_2$T$_x$ MXene[30]. The inset in Fig. 1d shows spectra for the acidic solutions with pure water

subtracted. The so-called acid continuum between 1700-3000 cm$^{-1}$ in the bulk case is featureless, apart from the bands at 1760 cm$^{-1}$ and 3000 cm$^{-1}$, which are attributed to stronger H-bonding occurring in the presence of hydrated protons[11]. In the MXene-confined acid solution, distinct bands are discernible at 1830 cm$^{-1}$, 2000 cm$^{-1}$, 2260 cm$^{-1}$ and 2520 cm$^{-1}$ as opposed to the bulk water which appears as a continuum. These bands correspond well with the proton features recorded on the dry Ti$_3$C$_2$T$_x$ film (Fig. 1c).

### *Operando* FTIR spectroscopy of protons intercalated in Ti$_3$C$_2$T$_x$ MXenes

*Operando* FTIR spectra were then recorded during electrochemical cycling in 0.1 M and 0.5 M H$_2$SO$_4$ at a scan rate of 2 mV/s (Fig. 2a, b). Multiple successive CVs were recorded (Supplementary Fig. 4) showing minor hysteresis, but overall the response is very stable, and the spectroscopic data recorded during different cycles is very reproducible. When negative polarization occurs, dramatic changes in the spectra are observed which are decomposed into 7 main peaks (Fig. 2c, d). Difference spectra are shown here for clarity but the full spectra are also available in Supplementary Fig. 5 and show that potential-dependent changes are observable even without subtractions. First,

the loss of neutral water within the $Ti_3C_2T_x$ film during proton intercalation is concluded from the decrease of the water bending mode at 1645 cm$^{-1}$. The increasing peaks 1 (-1920 cm$^{-1}$), 2 (-2080 cm$^{-1}$), 3 (-2380 cm$^{-1}$) and 4 (-2840 cm$^{-1}$) are attributed to hydrated protons and are not observed when the same measurement is performed using a neutral electrolyte (0.1 M $Li_2SO_4$, Supplementary Note 2 and Supplementary Fig. 6). In $D_2SO_4$, a clear isotopic shift of features 1-4 is observed (Supplementary Note 3 and Supplementary Fig. 7), confirming that these bands are associated with intercalated protons. Significant changes in the O−H stretching mode region (*ca.* 3000-3700 cm$^{-1}$) are also detected, assigned primarily to vibrations from water molecules located beyond the first excess proton hydration shell. A positive band that shifts from 3300 cm$^{-1}$ to 3220 cm$^{-1}$ (peak 5) is accompanied by a distinct negative band at 3530 cm$^{-1}$ (peak 6). These two bands together show that neutral water is eliminated within the $Ti_3C_2T_x$ interlayer spacing due to proton intercalation, and that the remaining water molecules experience stronger H-bonding. This observation, also performed on water trapped in graphene nanobubbles at high temperature[31], has previously been assigned to an increase of the water density in the confined environment[32]. The increase of the band at 3630 cm$^{-1}$ (peak 7) from dangling O−H bonds is also observed in a neutral electrolyte and therefore not specific to proton intercalation (Supplementary Fig. 6). As the scan direction is reversed and the potential cycled back towards 0 V, all absorbance bands decrease in area until no features are seen in the spectrum at 0 V showing a good reversibility of the proton intercalation process.

The spectral evolution is similar in both concentrations, with some notable exceptions (Fig. 2). Peaks 1 and 2 are significantly more prominent in the 0.5 M solution, whereas peak 3 is only observed in 0.1 M $H_2SO_4$, because it does not evolve with potential in 0.5 M $H_2SO_4$ (it is clearly present at this concentration before electrochemical cycling as seen in the inset of Fig. 1d). The larger negative area of peak 6, together with the decrease of the water bending mode at 1645 cm$^{-1}$ for the 0.5 M acid compared to the 0.1 M acid, suggests that a larger amount of neutral water is lost upon cycling with increasing acid concentration. Overall, a concentration dependence is observed for peaks 1–3, whereas the other peak areas are remarkably consistent in both concentrations. Comparing the peak fit results for 0.1 and 0.5 M acid concentrations, we interpret the evolution of the IR features with potential in terms of two parallel processes: (i) water reordering toward stronger H-bonding due to polarization at the MXene-electrolyte interface and (ii) proton intercalation and/or surface redox reactions. Process (i) is independent of proton concentration and is related to changes in water stretching modes as evidenced by peaks 4 and 5, the areas of which are nearly identical at both acid concentrations. Also, peak 7, attributed to water molecules with a dangling OH bond[6,28] appears at almost the same frequency and intensity in both, acidic and neutral electrolytes (Supplementary Fig. 6c, d). Peak 5, related to more coordinated water molecules, is more intense in 0.1 M $Li_2SO_4$ and is split into two components, reflecting an additional process of water co-intercalation with Li$^+$. Process (ii) is dependent on the proton concentration and gives rise to peaks 1-3 and loss of peak 6. Peaks 1-3 are associated with the diffusion of protons within the MXene interlayers, which replaces the weakly bonded neutral water (peak 5).

Some peaks exhibit considerable hysteresis in their integrated areas, which may be related to the state of protonation of the $Ti_3C_2$ surface and how that affects the way protons are hydrated in the interlayer space. As can be observed by visual inspection of the CV curve in Fig. 2f, the surface redox reaction that takes place at the $Ti_3C_2$ electrode deviates from the ideal, where the anodic and the cathodic peak potentials would be symmetric. The $Ti_3C_2$ film was optimized for IR study and the dropcasting method produced a film with a higher resistance compared to conventional supercapacitor electrode films.

The peak frequencies are summarized in Supplementary Fig. 8 for both concentrations. All peaks appear at identical or near-identical frequencies in both concentrations. Peaks 1, 2, 6 and 7 do not shift at all during potential cycling, whereas peaks 3, 4 and 5 shift to lower wavenumber with more negative potential. From the cyclic voltammograms (CVs) shown in Fig. 2f, we estimate that -2.3 and 2.6 protons per 12 MXene formula units are intercalated at the most negative potential in 0.1 M and 0.5 M $H_2SO_4$, respectively (see Supplementary Note 4 for details).

## Vibrational modes of intercalated protons

Density Functional Theory (DFT) calculations were performed to clarify the vibrational structure of the new bands in the region 1800-2400 cm$^{-1}$ associated with hydrated protons confined in the MXene interlayers. The absorbance spectra were calculated for two layers of water in the MXene interlayer space of ~5.1 Å and the number of protons was varied from zero to three protons per 12 MXene formula units (Fig. 3a). The calculated spectra were compared to experimental spectra recorded *operando* at 0 V and −0.6 V, the potentials that correspond to the lowest and highest number of protons, respectively, within the $Ti_3C_2T_x$ film. In the case of confined protons, three peaks were found in the calculated spectra that closely agree with peaks 1-3 in the experimental spectra in 0.1 M $H_2SO_4$ (Fig. 3b). The normal modes corresponding to these peaks all exhibit stretching and bending characteristics of the central hydronium ion mixed with bending and/or stretching of three water molecules in the first hydration shell. The most representative normal modes are shown in Fig. 3c and further details on the other ones are available in Supplementary Tables 2–4. Peaks 1–2 do not appear in the calculated spectrum of one proton in bulk water (Supplementary Fig. 10b), confirming that the specific vibrational modes are only present when the hydrated protons are confined within the $Ti_3C_2T_x$ interlayers.

The classification of proton hydration structures as Eigen or Zundel is often based on the proton-sharing parameter, δ, which describes the difference in the distance of the shared excess proton from the O atoms of the two nearest water molecules (for more details see Methods). δ = 0 Å for an ideal Zundel cation, whereas configurations with δ > 0.1 Å are classified as Eigen states. The simulated snapshots show practically no Zundel character for any normal modes contributing to peaks 1 and 2; instead, these peaks arise predominantly from Eigen modes. For peaks 1-3, both δ and peak frequency decrease in the order 3 > 2 > 1 (Table 1). Smaller δ has previously been shown to result in lower vibrational frequency[33]. Asymmetrically hydrated Eigen cations (one water molecule removed from or one water molecule added to $H_9O_4^+$) exhibit redshifts in the Eigen core frequency and present bands close to peaks 1 and 2 observed here[7]. Similarly, hydrated protons confined inside reverse micelles were found to exist predominantly in asymmetric Eigen-like hydration structures, with one weaker H-bond and two stronger H-bonds formed by the central hydronium ion[5]. Moreover, vibrational sum frequency spectroscopy measurements that are sensitive to interfacial modes show that hydrated protons at a charged liquid-vapour interface have a preferred orientation and strongly favour an Eigen-like configuration[34]. We do not observe a peak at 1750 cm$^{-1}$, which is prominent in difference spectra of bulk acid solutions and widely attributed to the bending mode of a Zundel cation[5,11,35], further supporting our DFT results and the assignment of peaks 1-3 to Eigen modes. The assignment of the experimental features based on these DFT calculations is summarized in Table 1.

A surface redox reaction is also expected to occur according to the equation:

$$Ti_3C_2O_2 + nH^+ + ne^- \rightleftharpoons Ti_3C_2O_{2-n}(OH)_n \qquad (1)$$

However, we are currently unable to definitively assign a particular absorption band to the surface hydroxyl groups. DFT calculations predict the stretching mode of the surface -OH to appear at ca.

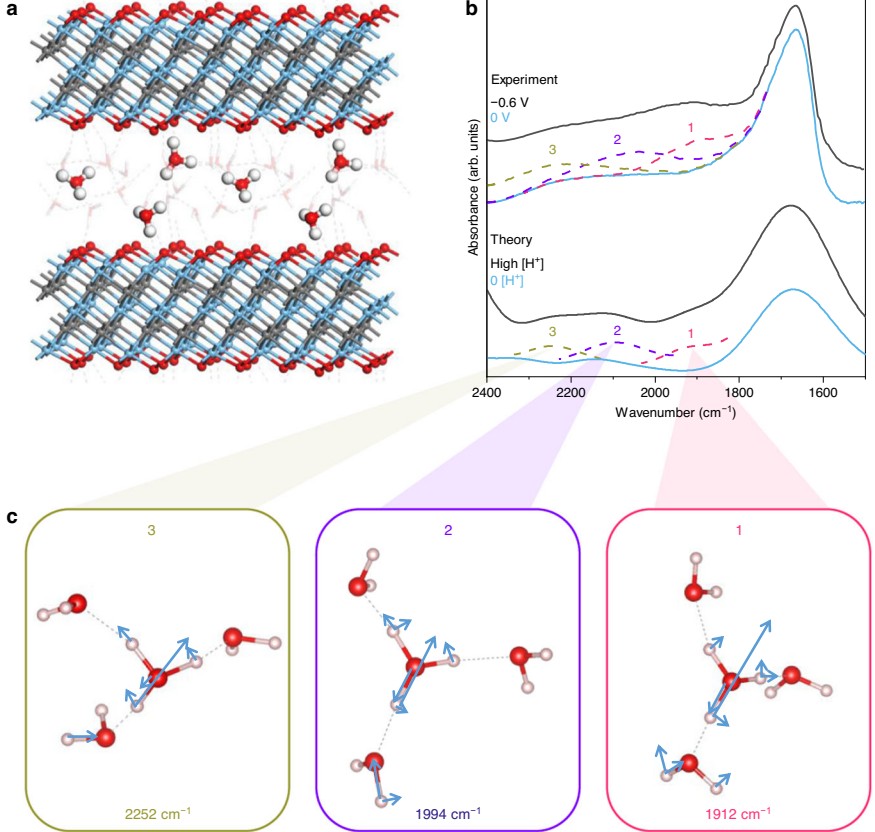

**Fig. 3 | Vibrational signature of hydrated protons in the interlayer space of Ti$_3$C$_2$T$_x$ MXene. a** The structure of three protons confined within the MXene interlayer space. Two unit cells are shown. **b** Top: *Operando* FTIR spectra at 0 V (blue) and −0.6 V (black) in 0.1 M H$_2$SO$_4$ referenced to the bare graphene-covered Si IRE represent low and high proton content, respectively. The dotted green, purple and red lines show peak components 1, 2 and 3 superimposed onto the spectrum at 0 V. Bottom: Calculated spectra of two layers of water in the MXene interlayer space with 0 protons (blue) and 1–3 protons (black). The dashed green, purple and red lines show the peak components. **c** The main vibrational normal modes contributing to the peaks are highlighted in corresponding colours, indicating the important contribution from the special pair (the hydronium ion with its closest neighbouring water) in the Eigen state (the proton-sharing parameter, δ, >0.1 Å, see Methods).

3730 cm$^{-1}$ in vacuum[25]; however, in the presence of hydrogen-bonding to water, this mode is expected to redshift. Surface -OH groups may contribute to peak 6 at 3630 cm$^{-1}$ but this assignment is not straightforward since this peak is also seen to increase in size in neutral electrolyte under a more negative potential. In our DFT molecular dynamics simulations, we can see that the surface OH groups are highly dynamic, with fast and reversible deprotonation-protonation in the local environment including the hydrogen bond network in confinement. We have taken about 10 instantaneous snapshots and analysed the normal models of the surface OH groups and found that they vary from 1700 to 3300 cm$^{-1}$ and are sensitive to the O−H distance. Hence, we think that the surface O−H vibration

modes are likely buried in the background. NMR[36] and Raman[25] spectroscopies, which can access other Ti-OH vibrational modes at low wavenumbers (200-600 cm$^{-1}$), may be more appropriate to resolve the Ti-OH contribution. X-ray absorption spectroscopy has proved useful in revealing the change in the Ti chemical bonding environment induced by hydroxylation and redox reaction of titanium carbide MXenes[19,37,38].

## 2D confinement of protons in Ti$_3$C$_2$T$_x$ MXene interlayer

To find out the structural origin responsible for the vibrational modes of hydrated protons in confinement, we compared the H-bonding network of the hydrated proton in confinement with that of a hydrated proton in bulk water. For a fully solvated hydronium ion, the three hydrogen atoms each donate a H-bond to a water molecule (labelled H$_D$), and the oxygen in the hydronium accepts one H-bond from a neighbouring water molecule (labelled O$_A$). These interactions are highlighted in the inset of Fig. 4a with blue and orange dashed lines, respectively. The donated H-bonds of the hydronium ion all have a length of ~1.6 Å and involve three water molecules for both the bulk and the confined systems. On the other hand, the accepted H-bond of the confined hydronium ion is significantly influenced by confinement. At the H-bond cutoff distance of 2.5 Å[32,39], the proton coordination number drops from 3.5 in the bulk to 3.0 in the confined interlayer space at both concentrations. This decrease of the hydronium coordination number matches well with the main normal modes of the confined protons, which all involved three water molecules. Recent theoretical and experimental studies have found that the protonated

## Table 1 | Peak assignments for operando FTIR spectra of Ti$_3$C$_2$T$_x$ MXene in dilute H$_2$SO$_4$ electrolyte

| Peak # | Mean frequency (cm$^{-1}$) | Assignment | Ref. |
|---|---|---|---|
| 1 | 1923 | $\nu_{O-H}$, Eigen core (δ = 0.25 Å) | This work |
| 2 | 2078 | $\nu_{O-H}$, Eigen core (δ = 0.32 Å) | This work |
| 3 | 2385 | $\nu_{O-H}$, Eigen core (δ = 0.37 Å) | This work |
| 4 | 2839 | $\nu_{O-H}$, Eigen core (2500-2650 cm$^{-1}$) and Zundel flanking water (3185 cm$^{-1}$) | 7,11,50 |
| 5 | 3262 | $\nu_{O-H}$, strongly H-bonded water | 11,51 |
| 6 | 3533 | $\nu_{O-H}$, weakly H-bonded water | 51 |
| 7 | 3621 | $\nu_{O-H}$, dangling OH of interfacial water | 3,4 |

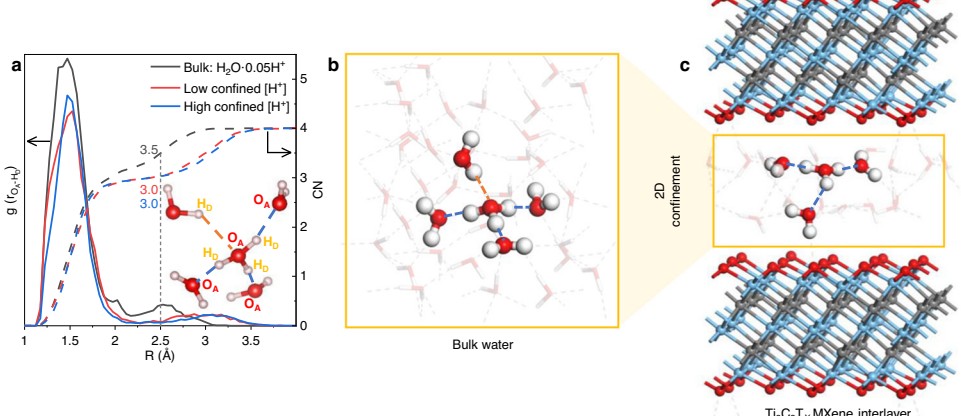

**Fig. 4 | Hydrogen bonding in 2D confinement. a** Radial distribution function of non-covalently bonded $O_A$-$H_D$ distances between a hydronium ion and its four nearest-neighbour water molecules (solid lines) and the resulting coordination numbers of the hydronium ion (dashed lines) in bulk water with 0.05 H⁺ per $H_2O$ (black), MXene-confined water at low [H⁺] ($Ti_3C_2O_2 \cdot 1.42H_2O \cdot 0.08H$, red), and MXene confined water at high [H⁺] ($Ti_3C_2O_2 \cdot 1.42H_2O \cdot 0.25H$, blue). The vertical dotted line represents the hydrogen bond length cutoff at 2.5 Å. The coordination numbers at this O–H distance are marked in the figure. Inset: Hydronium ion with 4 nearest water molecules; the H-bonds are marked by dashed lines. **b** Fully H-bonded hydronium ion in bulk water. **c** Schematic representation of the effect of 2D confinement on proton hydration.

water cluster H⁺(H₂O)₂₁, referred to as a magic number cluster due to its stability, exhibits IR bands in the 1800-2400 cm⁻¹ region[6,40]. In the most stable conformer of the H⁺(H₂O)₂₁ cluster, the excess proton is found in an Eigen configuration with three donated H-bonds but no accepted H-bond, similar to the H-bonding structure found in our DFT calculations. The loss of the accepted H-bond is therefore the main change in the hydration of the protons which is observed when the protons are confined within a two-dimensional environment in MXene, as represented schematically in Fig. 4b, c.

In summary, we have found that there are discrete vibrational modes of protons that are confined within the interlayer spaces $Ti_3C_2T_x$ using *operando* ATR-FTIR. These vibrational modes differ significantly from the continuum band that is observed for protons in bulk water. The major increase of these bands at negative potentials provides spectroscopic evidence of the intercalation of protons within $Ti_3C_2T_x$ layers during electrochemical cycling. At this stage, the associated vibrational modes are interpreted as a consequence of the reduction of the coordination number of hydronium ions resulting from 2D confinement between the layers of $Ti_3C_2T_x$ and possess an Eigen-like character. A strengthening of the overall water H-bonding is also observable for high proton concentrations. The peculiar proton hydration occurring under 2D confinement may correlate with the high pseudocapacitive electrochemical energy storage performance that has been reported not only for MXenes[19,20] but also for other 2D and layered materials[41]. Going forward, this experimental method can be applied to probe the intercalation of any type of ion, molecule or electrolyte within the interlayer spacing of MXenes, thereby offering new opportunities for the characterization of chemical species under 2D confinement.

## Methods

### MXene synthesis
$Ti_3C_2T_x$ MXene was synthesized according to the procedure described in[26]. By including excess aluminium during synthesis of the $Ti_3AlC_2$ MAX phase precursor, single- and few-layer $Ti_3C_2T_x$ MXene was obtained with improved stoichiometry, resistance to oxidation and increased electrical conductivity. Briefly, the MAX phase was synthesized from TiC, Ti, and Al powders at 1380 °C under a constant argon flow. The washed, dried, and sieved $Ti_3AlC_2$ precursor was etched in a mixture of HCl and HF to produce multilayered MXene, which was then delaminated by dispersing the multilayer MXene in a LiCl solution to

obtain single- and few-layer $Ti_3C_2T_x$ MXene. A stock aqueous suspension at a concentration of 5.9 mg/ml was stored under argon in a sealed bottle and fresh aliquots were drawn on the day of the experiments.

### MXene film preparation
All electrolytes and $Ti_3C_2T_x$ suspensions were made with doubly deionized water (Millipore, resistivity 18.2 MΩ • cm). $Li_2SO_4$ and $H_2SO_4$ (Sigma-Aldrich) were used as received. All electrolytes were deoxygenated by bubbling with nitrogen for 30 min prior to experiments. A $Ti_3C_2T_x$ stock suspension was diluted to 1 mg/ml, 125 μl was pipetted onto a graphene-covered Si wafer (*vide infra*) and the droplet was allowed to dry at room temperature. Some manual manipulation of the drying droplet was necessary to ensure that the infrared beam probed an area fully covered by MXene. The resulting film was ca. 600 nm thick. Measurements were also made with thinner films by reducing the amount of $Ti_3C_2T_x$ MXene suspension pipetted onto the wafer.

### Electrochemical measurements
Electrochemical measurements were performed in a three-electrode spectroelectrochemical cell designed and built in-house (Supplementary Fig. 1). Electrical contact to the $Ti_3C_2T_x$ working electrode was made through the graphene film, although measurements were also performed without the graphene layer with very similar electrochemical results. A coiled Pt wire and an Ag wire were used as the counter and quasi-reference electrodes, respectively. Prior to use, the Ag wire was anodized at 50 mV vs. OCP in 0.1 M HCl. The potential was controlled with a Bio-Logic SP-200 potentiostat running EC-Lab software. With all electrodes mounted, the cell was filled with electrolyte, sealed, mounted into the infrared spectrometer and the spectrometer was evacuated until the pressure in the sample chamber reached 0.8 mbar (typically ca. 1 h). After this equilibration time, the open-circuit potential (OCP) of the system was recorded and used as the starting potential for the CV. Each CV was recorded between 0 V and −0.6 V with a scan rate of 2 mV/s. Several cycles were performed to verify the stability of the MXene electrode.

### FTIR measurements
The measurements were carried out at the IRIS beamline at the BESSY II electron storage ring operated by the Helmholtz-Zentrum Berlin für Materialen und Energie[42]. A conventional internal broadband IR source

was used for the measurements. FTIR spectra were collected with Bruker 70 v spectrometer equipped with a liquid nitrogen cooled MCT detector. The internal reflection element (IRE) was a microstructured Si wafer (Irubis) covered by a monolayer film of graphene (Graphenea) transferred as per instructions. A fresh graphene-covered wafer was prepared for each measurement. The graphene layer improved adhesion between the MXene film and the IRE, leading to a higher sensitivity to interlayered water layers and more reversible spectra during electrochemical cycling. The optical accessory was designed and built in-house to accommodate the microstructured Si wafer as the IRE and provides an angle of incidence of 28.74°. Each spectrum consisted of 128 scans and took ~30 s to record, with a 20-s waiting time between spectra resulting in one spectrum every 50 s. With a cyclic voltammetry scan rate of 2 mV/s, this equates to one spectrum per 0.1 V. The timing was manually synchronized with the cyclic voltammetry such that the midpoint of each spectrum collection coincided with potential = 0.1n V, where $n$ = integer. For example, an FTIR spectrum, the collection of which started when the potential was −0.17 V and finished when the potential was −0.23 V, is designated as the spectrum at −0.2 V. A linear or cubic baseline was subtracted from spectra.

### DFT calculations

All the density functional theory (DFT) calculations were performed by Vienna Ab initio Simulation Package (VASP)[43]. The nuclei-electron interaction was treated by projector augmented wave (PAW) potential[44]. The Perdew-Burke-Ernzerhof (PBE) version of generalized gradient approximation (GGA) was adopted for exchange-correlation functional[45]. The energy cutoff of the plane wave basis sets was set to 500 eV. The optB88-vdW functional was adopted to account for the van der Waals interaction[46]. Supercells containing 12 formula units of $Ti_3C_2O_2$ were used to model MXene. 17 water molecules with different numbers of protons (0, 1, 2 and 3) were randomly placed in between MXene layers. The convergence criteria were set to $10^{-5}$ eV in energy and 0.01 eV/Å in force.

To sample the configurations of intercalated water/proton, first principles molecular dynamics simulations (FPMD) were performed at 300 K for 10 ps with a timestep of 0.5 fs, in the canonical ensemble (NVT) using the Nose-thermostat[47]. From each trajectory, snapshots at every 0.5 ps from 2 ps to 10 ps were then selected for vibrational analysis. For each snapshot, density functional perturbation theory (DFPT) was applied to obtain the normal modes $[e_\beta(l)]$ and the Born effective charges $(Z^*_{\alpha\beta})$. The infrared intensity of each normal mode was calculated as[48]

$$I = \Sigma_\alpha [\Sigma_l \Sigma_\beta Z^*_{\alpha\beta} e_\beta(l)]^2 \qquad (2)$$

To simulate the whole IR spectrum, the peak associated with each normal mode was then broadened by a Gaussian function with a width of 60 cm$^{-1}$. The final IR spectrum was then obtained by averaging over the selected snapshots. The frequency axis was scaled by a factor of 1.017 and 1.039 such that the water bending mode was centered at 1635 cm$^{-1}$ and 1665 cm$^{-1}$ in the bulk and confined spectra, respectively. To mimic the heterogeneity of proton concentration in experimental measurement, the simulated spectrum in Fig. 3B is the averaged spectrum of the two simulated compositions: $Ti_3C_2O_2 \cdot 1.42H_2O \cdot 0.08H$ and $Ti_3C_2O_2 \cdot 1.42H_2O \cdot 0.25H$.

### Proton-sharing parameter

The hydrated proton is often described as either a Zundel or Eigen cation. In the Zundel cation, one excess proton is equally shared by two water molecules with a further four water molecules in the first hydration shell, whereas in the Eigen cation the excess proton is more closely associated with one water molecule, surrounded by a hydration shell of three additional water molecules. The two water molecules sharing the excess proton through an $O_D$——$H$—$O_A$ hydrogen bond are

referred to as the special pair, and $O_D$ ($O_A$) refers to the oxygen from the H-bond donor (acceptor) molecule. The classification of hydration structures is defined through the proton-sharing parameter δ:

$$\delta = |r_{O_DH} - r_{O_AH}| \qquad (3)$$

where $r$ is the distance between $O_i$ and H. Special pair configurations where δ < 0.1 Å are classified as Zundel and δ > 0.1 Å as Eigen[33]. It is important to note that this cutoff value is arbitrary; other classification methods can be found in the literature[49]. A perfectly symmetric Zundel cation would have δ = 0 Å and a perfectly symmetric Eigen cation would have δ = 0.53 Å[41].

## Data availability

The FTIR data generated in this study has been deposited in the Zenodo database under accession code https://doi.org/10.5281/zenodo.7600123. Further data is available upon request to the authors.

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

## Acknowledgements

The authors would like to acknowledge financial support by Volkswagen Foundation Freigeist Fellowship 89592 (M.L., T.P.), European Research Council (ERC) under the European Union's Horizon 2020 research and innovation programme, grant 947852 (M.L., T.P.), and Fluid Interface Reactions, Structures and Transport (FIRST) Center, an Energy Frontier Research Center funded by the US Department of Energy, Office of Science, and Office of Basic Energy Sciences (Y.S., T.S.M., D.J., Y.G.). The authors would like to thank the Helmholtz-Zentrum Berlin für Materialen und Energie for the use of the IRIS beamline facilities, and Martin Künsting for the illustration in Fig. 1a.

## Author contributions

M.L., Y.G. and T.P. formulated the idea and conceived the experiments. D.J. conceived the DFT calculations. M.L. performed the FTIR measurements. Y.S. performed the DFT calculations under the supervision of D.J. M.L., U.S., L.P. and T.P. designed custom equipment for the experiments. T.S.M. synthesized and characterized the $Ti_3C_2T_x$ MXene under the supervision of Y.G., M.L. and Y.S. analysed the data. All authors contributed to the interpretation of the data. M.L., Y.S., D.J., Y.G. and T.P. worked on the visualization of the data. M.L. and T.P. wrote the original draft. All authors reviewed, edited, and approved the manuscript.

## Funding

## Competing interests

The authors declare no competing interests.
