## [Peer Review File · Nature Communications]

Vibrational signature of hydrated protons confined in MXene interlayersREVIEWER COMMENTS

Reviewer #1 (Remarks to the Author):

The authors investigate the ATR-FTIR spectra of MXene interlayers in the presence of water and sulfuric acid solutions and identify spectral features associated with the hydrated proton during electrochemical cycling conditions. The signal was enhanced at the lowest potentials measured (i.e. -0.6 V) when the proton concentration in the interlayers is expected to be maximum. The authors claim they detect discrete vibrational modes that differ "significantly" from the continuum band observed in the bulk. They also complement their operando FTIR studies with DFT calculations. The work is generally interesting, but the authors appear to overstate their conclusions.

Primarily, the vibrational modes detected are not "significantly" different from those observed in the bulk. The difference spectra presented in Figures 1 and 2 show very much the same features are those reported previously for bulk acid solutions at slightly higher concentrations (see, for instance, Tokmakoff and coworkers (ref #10 and #11 in the manuscript), JACS 2010, 132, 1484, and/or JPCL 2017, 8, 5246). The results should be contrasted with studies where either due to confinement (PCCP 2020, 22, 21334; included the manuscript as Ref#5 but not cited for this reason) or the presence of a surface (Nature Comm. 2020, 11, 493) distinct features in the continuum band were observed.

Indeed, as shown in this work, the compelling advantage of the MXene interlayers is that when applying a negative potential, protons are absorbed, and their local concentration can be substantially enhanced. Still, like for the bulk case, reference spectral subtraction is required, as illustrated by the authors in their SI (Figure S4). Nonetheless, the higher local proton concentration in the MXene allows the detection of intriguing changes in the spectral response, yet, the interpretation appears to be partly incomplete. For example, Figure 2E shows how the different fitted peak areas change with the applied potential. Even though the number of bands chosen could be debatable, besides the obvious, it is not apparent what relevant information can be extracted from the analysis. The discussion should be further developed.

Another incomplete point is related to the Zundel-Eigen debate. The DFT results (models in Figures 3 and 4, and particularly in the SI) appear to show a marked preference for the Eigen species in MXene. However, this theoretical finding is eluded in the discussions and conclusions of the manuscript. Are the DFT results not trustworthy? This point should be further clarified.

Overall, I find this work appealing and potentially relevant for the general readership of Nature Communication. However, the text and interpretations should be further improved before considering for publication.

Reviewer #2 (Remarks to the Author):

In "Vibrational signature of hydrated protons confined in MXene interlayers", Lounasvuori and coworkers present a study purporting to fingerprint solvated proton between MXene layers, using operando FTIR spectroscopy and DFT calculations. The spectral signature of solvated protons is scientifically relevant but difficult to ascertain, due to the transient nature of system and their relatively low concentration in the bulk. This work then represents an important advance, not only in applying an interesting technique, but also in showing significant new physics in a 2d confined aqueous system. The experiments and theory are complementary and well-connected and the interpretations are sound and well supported. Overall, I recommend publication of this work, provided that the authors address the following:

1. Were the systems cycled multiple time? Was there any hysteresis observed in the CV scans in figure 2
2. Is there anything special about Mxenes in the generation of these solvated proton structures? Would similar peaks be possible in other 2d systems (like graphene/hexagonal BN)?
3. The calculated spectrum is an average of 16(32?) individual snapshot. How much variability is there in these spectrums? A plot of should be provided in the SI so that the reader can assess convergence.
4. Since the bulk solvated proton is missing these features, it would appear that if the calculations and measurements are consistent, their approach is extremely surface sensitive. Can the authors comment on the probability distribution function for the specific hydrated proton structure as a function of distance from the interface from their AIMD simulations?
5. The authors attribute the changes in their operando spectrum as due to i) water reordering and ii) proton intercalation/surface redox reactions. The second point refers to two distinct processes: faradic and non-faradic. However, from the discussion in the outlook, the main mechanism/opportunities is assumed to be from non-faradic capacitive changes. The authors are encouraged to expand on this further. Is there any way that the calculations can be used to probe the relative contributions of faradic processes?

Minor points:

The statement: "The space between clusters, which are limited to ca. 21 water molecules (6), and bulk water remains largely unexplored" is unclear

The statement: "Experimental spectra recorded operando at 0 V and -0.6 V, the potentials that correspond to the lowest and highest number of protons, respectively, within the $\text{Ti}_3\text{C}_2\text{Tx}$ film" is missing a word or two

REVIEWER COMMENTS

Reviewer #1 (Remarks to the Author):

The authors investigate the ATR-FTIR spectra of MXene interlayers in the presence of water and sulfuric acid solutions and identify spectral features associated with the hydrated proton during electrochemical cycling conditions. The signal was enhanced at the lowest potentials measured (i.e. -0.6 V) when the proton concentration in the interlayers is expected to be maximum. The authors claim they detect discrete vibrational modes that differ “significantly” from the continuum band observed in the bulk. They also complement their operando FTIR studies with DFT calculations. The work is generally interesting, but the authors appear to overstate their conclusions.

Primarily, the vibrational modes detected are not “significantly” different from those observed in the bulk. The difference spectra presented in Figures 1 and 2 show very much the same features are those reported previously for bulk acid solutions at slightly higher concentrations (see, for instance, Tokmakoff and coworkers (ref #10 and #11 in the manuscript), JACS 2010, 132, 1484, and/or JPCL 2017, 8, 5246). The results should be contrasted with studies where either due to confinement (PCCP 2020, 22, 21334; included the manuscript as Ref#5 but not cited for this reason) or the presence of a surface (Nature Comm. 2020, 11, 493) distinct features in the continuum band were observed.

Response: The most distinct difference in our spectra compared to bulk spectra is the absence of the Zundel bending mode at 1750 cm^{-1} . Instead, we observe a band at 1923 cm^{-1} , which is assigned to a stretching mode of the Eigen core based on the DFT calculations. We accept the Reviewer’s assessment that this does not amount to a significant difference and have removed the word “significantly” from the text.

We agree with the Reviewer’s suggestion to compare our results to other studies where distinct features in the continuum band are observed. In fact, there is already a section in the supporting information where we do this and we already refer to the first of the suggested papers (PCCP 2020, 22, 21334) in the supporting information when discussing the peak fitting of our data:

Asymmetrically hydrated Eigen cations (one water molecule removed from or one water molecule added to H_9O_4^+) exhibit redshifts in the Eigen core frequency and present bands close to peaks 1 and 2 observed here (7). Similarly, hydrated protons inside reverse micelles were found to exist predominantly in asymmetric hydration structures, with one weaker H-bond and two stronger H-bonds formed by the central hydronium ion (8).

In light of the Reviewer’s comments, we have extended the discussion, moved the above passage to the main manuscript from the supporting information, and added discussion citing the second suggested paper (Nature Comm. 2020, 11, 493).

Amended text:

The classification of proton hydration structures as Eigen or Zundel is often based on the proton-sharing parameter, δ , which describes the difference in the distance of the shared excess proton from the O atoms of the two nearest water molecules (for more details see Methods). $\delta = 0$ for an ideal Zundel cation, whereas configurations with $\delta > 0.1\text{ \AA}$ are

classified as Eigen states. The simulated snapshots show practically no Zundel character for any normal modes contributing to peaks 1 and 2; instead, these peaks arise predominantly from Eigen modes. For peaks 1-3, both δ and peak frequency decrease in the order $3 > 2 > 1$. Smaller δ has previously been shown to result in lower vibrational frequency (33). Asymmetrically hydrated Eigen cations (one water molecule removed from or one water molecule added to $H_9O_4^+$) exhibit redshifts in the Eigen core frequency and present bands close to peaks 1 and 2 observed here (7). Similarly, hydrated protons confined inside reverse micelles were found to exist predominantly in asymmetric Eigen-like hydration structures, with one weaker H-bond and two stronger H-bonds formed by the central hydronium ion (5). Moreover, vibrational sum frequency spectroscopy measurements that are sensitive to interfacial modes show that hydrated protons at a charged liquid-vapor interface have a preferred orientation and strongly favor an Eigen-like configuration (34). We do not observe a peak at 1750 cm^{-1} , which is prominent in difference spectra of bulk acid solutions and widely attributed to the bending mode of a Zundel cation (5, 11, 35), further supporting our DFT results and the assignment of peaks 1-3 to Eigen modes. The assignment of the experimental features based on these DFT calculations is summarized in the Table 1.

Table 1. Peak assignments for operando FTIR spectra of $Ti_3C_2T_x$ MXene in dilute H_2SO_4 electrolyte.

Peak #	Mean frequency (cm^{-1})	Assignment	Ref.
1	1923	ν_{O-H} , Eigen core ($\delta = 0.25\text{ \AA}$)	This work
2	2078	ν_{O-H} , Eigen core ($\delta = 0.32\text{ \AA}$)	This work
3	2385	ν_{O-H} , Eigen core ($\delta = 0.37\text{ \AA}$)	This work
4	2839	ν_{O-H} , Eigen core (2500-2650 cm^{-1}) and Zundel flanking water (3185 cm^{-1})	(7, 11, 36)
5	3262	ν_{O-H} , strongly H-bonded water	(11, 37)
6	3533	ν_{O-H} , weakly H-bonded water	(37)
7	3621	ν_{O-H} , dangling OH of interfacial water	(3, 4)

Indeed, as shown in this work, the compelling advantage of the MXene interlayers is that when applying a negative potential, protons are absorbed, and their local concentration can be substantially enhanced. Still, like for the bulk case, reference spectral subtraction is required, as illustrated by the authors in their SI (Figure S4). Nonetheless, the higher local proton concentration in the MXene allows the detection of intriguing changes in the spectral response, yet, the interpretation appears to be partly incomplete. For example, Figure 2E shows how the different fitted peak areas change with the applied potential. Even though the number of bands chosen could be debatable, besides the obvious, it is not apparent what relevant information can be extracted from the analysis. The discussion should be further developed.

Response: Regarding the Reviewer's comment on the need to present difference spectra in order to see the vibrational modes arising from hydrated protons, we would like to point out that we do, in fact, see clear changes in the spectra presented in Figure 1D even without subtraction, at 0.5M electrolyte concentration. We agree that the concentration in Figure S4

(new Figure S5) are related to much higher proton concentration and cannot be fairly compared to bulk electrolyte.

The number of components in the peak fit was decided based on second derivative analysis. Figure R1 shows the second derivative of the spectrum recorded at -0.6 V, and we can clearly discern three components in the region 2500 - 1750 cm^{-1} .

Figure R1. FTIR difference spectrum recorded at -0.6 V during electrochemical cycling of Ti_3C_2 electrode in 0.1 M H_2SO_4 (blue) and the second derivative of said spectrum (orange). The dotted vertical lines mark the three components of the peak fit labelled 1-3.

We discuss the peak fit in the main manuscript and expand on it in Supporting Information. The main information extracted from the analysis is that peaks 1-3 are concentration-dependent, whereas the other components are nearly identical in both concentrations. From this, we conclude that we have two processes occurring in parallel, as discussed in the supporting information.

We have moved the discussion from the supporting information into the main manuscript and developed it further.

Amended text:

Comparing the results for 0.1 and 0.5 M acid concentrations, we interpret the evolution of the IR features with potential in terms of two parallel processes: i) water reordering toward higher H-bonded water due to polarization at the MXene-electrolyte interface and ii) proton intercalation and/or surface redox reactions. Process i) is independent of proton concentration and is related to changes in water stretching modes as evidence by peaks 4 and 5, the areas of which are nearly identical in both acid concentrations. Also, peak 7, attributed to water molecules with a dangling OH bond (3, 4) appears at almost the same frequency and intensity in both acidic and neutral electrolyte (Fig. S5C, D). Peak 5, related to more coordinated water molecules, is more intense in 0.1 M Li_2SO_4 and is split into two

components, reflecting an additional process of water co-intercalation with Li^+ . Process ii) is dependent on the proton concentration and gives rise to peaks 1-3 and loss of peak 6. Peaks 1-3 are associated with the diffusion of protons within the MXene interlayers, which replaces the weakly bonded neutral water (peak 5).

Some peaks exhibit considerable hysteresis in their integrated areas, which may be related to the state of protonation of the Ti_3C_2 surface and how that affects the way protons are hydrated in the interlayer space. As can be observed by visual inspection of the CV curve in Fig. 2F, the surface redox reaction that is expected to take place at the Ti_3C_2 electrode deviates from the ideal where the anodic and the cathodic peak potentials would be the same. The Ti_3C_2 film is not optimized for electrochemistry and the dropcasting method would introduce additional resistance to the film.

Another incomplete point is related to the Zundel-Eigen debate. The DFT results (models in Figures 3 and 4, and particularly in the SI) appear to show a marked preference for the Eigen species in MXene. However, this theoretical finding is eluded in the discussions and conclusions of the manuscript. Are the DFT results not trustworthy? This point should be further clarified.

Response: In the main manuscript we decided to focus on the reduced coordination number of the confined hydrated proton as this was deemed to be the most important finding from the DFT results. However, when discussing Figure 4, we do state the following:

In the most stable conformer of the $\text{H}^+(\text{H}_2\text{O})_{21}$ cluster, the excess proton is found in an Eigen configuration with three donated H-bonds but no accepted H-bond, similar to the H-bonding structure found in our DFT calculations.

In the supporting information, we discuss the Eigen character indicated by the DFT results and assign our peaks 1-3 to Eigen-like modes based on the DFT results.

Amended text: Please see response to Reviewer 1, point 1.

Overall, I find this work appealing and potentially relevant for the general readership of Nature Communication. However, the text and interpretations should be further improved before considering for publication.

Response: We thank the Reviewer for the careful evaluation of our work and their suggestions to improve the manuscript. We hope that the revised manuscript can now be considered for publication.

Reviewer #2 (Remarks to the Author):

In "Vibrational signature of hydrated protons confined in MXene interlayers", Lounasvuori and coworkers present a study purporting to fingerprint solvated proton between MXene layers, using operando FTIR spectroscopy and DFT calculations. The spectral signature of solvated protons is scientifically relevant but difficult to ascertain, due to the transient nature

of system and their relatively low concentration in the bulk. This work then represents an important advance, not only in applying a interesting technique, but also in showing significant new physics in a 2d confined aqueous system. The experiments and theory are complementary and well-connected and the interpretations are sound and well supported. Overall, I recommend publication of this work, provided that the authors address the following:

1. Were the systems cycled multiple time? Was there any hysteresis observed in the CV scans in figure 2

Response: The systems were cycled multiple times to ensure the stability. The electrochemical data recorded during the operando measurement in 0.1 M H₂SO₄ is presented in Fig. R2. Some minor hysteresis is observed but, overall, the response is very stable, and the spectroscopic data recorded during different cycles is very reproducible.

Previous studies have already shown that Ti₃C₂ is extremely stable in dilute sulfuric acid electrolytes (typical concentrations 1-3 M H₂SO₄) and thin film electrodes of Ti₃C₂ are routinely cycled over thousands of cycles (see, for example, DOI: 10.1038/nenergy.2017.105, DOI: 10.1039/C8NR01550C).

Figure R2. Cyclic voltammogram recorded in 0.1 M H₂SO₄ during the operando FTIR measurements presented in the manuscript.

2. Is there anything special about MXenes in the generation of these solvated proton structures? Would similar peaks be possible in other 2d systems (like graphene/hexagonal BN)?

Response: We did cite some examples of water and proton intercalation in graphene and hBN in the introduction (DOI: 10.1038/nature14295, DOI: 10.1126/science.aau6771, DOI: 10.1126/science.aat4191) but we indeed believe that MXenes are rather unique among 2D materials. They combine high conductivity with hydrophilic surfaces consisting of a large number of O/OH terminal groups (DOI: 10.1038/s41570-022-00384-8). While water can intercalate between pristine graphene or hBN sheets, it does not form hydrogen bonds with

the material surface. Moreover, most examples of water intercalated between graphene sheets involve intricate engineering of the structure (DOI: 10.1038/nature19363, DOI: 10.1126/science.aau6771) or disconnected water domains (DOI: 10.1038/nature14295), whereas the solution processability of Ti_3C_2 greatly simplifies the experimental steps. The high negative charge of MXene surface (also ensuring repulsion of anions) together with the possibility to apply negative potential without degrading the MXene (much higher negative potentials would be required to overcome the van der Waals attraction between graphene sheets) facilitate the insertion of quite high proton concentration which, to our knowledge, seems hardly possible with other 2D materials. That being said, we hope that our work will stimulate research efforts for IR characterization of protons and other ions intercalated in 2D materials and a detailed comparison with the results presented here for MXene would certainly provide new insights about confinement effects in 2D materials.

3. The calculated spectrum is an average of 16(32?) individual snapshot. How much variability is there in these spectrums? A plot of should be provided in the SI so that the reader can assess convergence.

Response: We have used 16 snapshots for each of the two different proton concentrations simulated, so there are 32 snapshots in total that have been averaged. The variations are shown for a typical concentration in the figure below for the most relevant regions (2400 to 1800 cm^{-1}), and one can indeed see the emergence of three main bands: 2240 , 2100 , and 1900 cm^{-1} . We have included this figure in the SI.

Figure R3. Simulated IR spectra for 16 snapshots from the AIMD simulation of a typical proton concentration ($\text{Ti}_3\text{C}_2\text{O}_2 \cdot 1.42\text{H}_2\text{O} \cdot 0.08\text{H}$); each line represents one snapshot.

4. Since the bulk solvated proton is missing these features, it would appear that if the calculations and measurements are consistent, their approach is extremely surface sensitive. Can the authors comment on the probability distribution function for the specific hydrated proton structure as a function of distance from the interface from their AIMD simulations?

Response: In our MXene-confined water/proton systems, the thickness of the water is about two molecular layers, which is based on the experimental measured c-lattice parameter during cycling (DOI: 10.1002/adfm.201902953); in other words, all the water molecules and solvated protons are at the interface, interacting with either the top or the bottom MXene surface. This is particularly interesting because we therefore do not need a high surface sensitivity – all probed protons and water molecules through the MXene film are in an interfacial region. As such, we do not see how the distribution function as a function of distance from the interface would be defined and how this would facilitate the discussion.

In our previous AIMD simulations of MXene-confined water/proton (DOI: 10.1021/acscami.9b18139), we have varied the water thickness from one layer to three layers and found that the proton behavior and solvation structure moves toward bulk-like in the three-layer case and, by extrapolation, would be bulk-like when the thickness is greater than four. As we never reach 4 water layers in the MXene interlayer spacing, it is not unexpected that we have a different behaviour from bulk protons in the confined environment.

5. The authors attribute the changes in their operando spectrum as due to i) water reordering and ii) proton intercalation/surface redox reactions. The second point refers to two distinct processes: faradic and non-faradic. However, from the discussion in the outlook, the main mechanism/opportunities is assumed to be from non-faradic capacitive changes. The authors are encouraged to expand on this further. Is there any way that the calculations can be used to probe the relative contributions of faradic processes?

Response: In the supporting information, we comment on the lack of definitive assignment of a surface hydroxyl group that would allow us to probe the contribution of the Faradaic process involving surface redox reactions on the Ti_3C_2 :

Some surface redox reactions are also expected to occur according to the equation:

Currently we are unable to definitively assign a particular absorption band to the surface hydroxyl groups. DFT calculations predict the stretching mode of the surface -OH to appear at ca. 3730 cm^{-1} in vacuum (5); however, in the presence of hydrogen-bonding to water this mode is expected to redshift. Surface -OH groups may contribute to peak 6 at 3630 cm^{-1} but this assignment is not straightforward since this peak is also seen to increase in size in neutral electrolyte with more negative potential.

In our DFT MD simulations, we can see that the surface OH groups are highly dynamic with fast, reversible deprotonation-protonation, as a result of the local environment, including the hydrogen bond network in confinement. We have taken about 10 instantaneous snapshots and analysed the normal modes of the surface OH groups and found that they vary from 1700 to 3300 cm^{-1} and are sensitive to the O-H distance. Hence, we think that the surface O-H vibration modes are likely buried in the background.

In our opinion, NMR (DOI: 10.1039/c6cp00330c) and Raman (DOI: 10.1039/c4cp05666c) spectroscopies, which can access other Ti-OH vibrational modes at low wavenumbers (200 - 600 cm^{-1}), are more appropriate to resolve Ti-OH contribution. X-ray absorption

spectroscopy (DOI: 10.1021/acsenergylett.0c01290) has proved useful in revealing the change in the oxidation state of Ti induced by the pseudocapacitive redox reaction.

We have moved the discussion into the main manuscript and expanded on it.

Amended text:

Some surface redox reactions are also expected to occur according to the equation:

Currently we are unable to definitively assign a particular absorption band to the surface hydroxyl groups. DFT calculations predict the stretching mode of the surface -OH to appear at ca. 3730 cm⁻¹ in vacuum (25); however, in the presence of hydrogen-bonding to water this mode is expected to redshift. Surface -OH groups may contribute to peak 6 at 3630 cm⁻¹ but this assignment is not straightforward since this peak is also seen to increase in size in neutral electrolyte with more negative potential. In our DFT MD simulations, we can see that the surface OH groups are highly dynamic with fast, reversible deprotonation-protonation, as a result of the local environment, including the hydrogen bond network in confinement. We have taken about 10 instantaneous snapshots and analysed the normal models of the surface OH groups and found that they vary from 1700 to 3300 cm⁻¹ and are sensitive to the O-H distance. Hence, we think that the surface O-H vibration modes are likely buried in the background. NMR (38) and Raman (25) spectroscopies, which can access other Ti-OH vibrational modes at low wavenumbers (200-600 cm⁻¹), may be more appropriate to resolve the Ti-OH contribution. X-ray absorption spectroscopy has proved useful in revealing the change in the Ti chemical bonding environment induced by hydroxylation and pseudocapacitive redox reaction (19, 39, 40).

Minor points:

The statement: "The space between clusters, which are limited to ca. 21 water molecules (6), and bulk water remains largely unexplored" is unclear

Response: This statement was replaced with: "*In contrast with clusters, which are limited to ca. 21 molecules (6), and bulk water, the vibrational modes of hydrated protons in 2D confinement remain largely unexplored.*"

The statement: "Experimental spectra recorded operando at 0 V and -0.6 V, the potentials that correspond to the lowest and highest number of protons, respectively, within the Ti₃C₂T_x film" is missing a word or two

Response: This statement was replaced with: "*The calculated spectra were compared with experimental spectra recorded operando at 0 V and -0.6 V, the potentials that correspond to the lowest and highest number of protons, respectively, within the Ti₃C₂T_x film.*"

REVIEWERS' COMMENTS

Reviewer #1 (Remarks to the Author):

The authors have duly addressed all points raised by the two reviewers. I recommend publication.

Reviewer #2 (Remarks to the Author):

The authors have satisfactorily addressed my concerns and in some cases provided new data to back up their previous discussion. I'm now comfortable recommending publication in Nat. Comm